# Mg^2+^ Ions Regulating 3WJ-PRNA to Construct Controllable RNA Nanoparticle Drug Delivery Platforms

**DOI:** 10.3390/pharmaceutics14071413

**Published:** 2022-07-06

**Authors:** Le Chen, Jingyuan Li

**Affiliations:** Zhejiang Province Key Laboratory of Quantum Technology and Device, School of Physics, Zhejiang University, Zheda Road 38, Hangzhou 310027, China; 3180103853@zju.edu.cn

**Keywords:** RNA nanotechnology, 3WJ-pRNA, thermal stability, Mg^2+^ ions, regulation mechanism

## Abstract

RNA nanotechnology has shown great progress over the past decade. Diverse controllable and multifunctional RNA nanoparticles have been developed for various applications in many areas. For example, RNA nanoparticles can participate in the construction of drug delivery nanoplatforms. Recently, a three-way junction packaging RNA (3WJ-pRNA) has been exploited for its characteristics of self-assembly and ultrahigh stability in many aspects. 3WJ-pRNA is the 3WJ part of bacteriophage φ29 pRNA and joins different components of φ29 as a linker element. In this work, we used all-atom MD simulation to study the thermal stability of 3WJ-pRNA and the underlying mechanisms. While 3WJ-pRNA can remain in its original structure without Mg^2+^ ions at room temperature, only Mg-bound 3WJ-pRNA still maintains its initial three-way junction structure at a higher temperature (*T* = 400 K). The Mg-free 3WJ-pRNA undergoes dramatic deformation under high temperature condition. The contribution of Mg ions can be largely attributed to the protective effect of two Mg clamps on the hydrogen bond and base stacking interactions in helices. Taken together, our results reveal the extraordinary thermal stability of 3WJ-pRNA, which can be regulated by Mg^2+^ ions. Comprehensive depictions of thermal stability of pRNA and the regulation mechanism are helpful for the further development of controllable RNA nanoparticle drug delivery platforms.

## 1. Introduction

There are increasing studies on the nanometer-scale architectures with their major frames composed of RNA molecules [1]. These architectures are called RNA nanoparticles and have shown a broad application prospect in many areas [2]. In biomedicine, for example, RNA nanoparticles can deliver therapeutic drugs in a targeted manner based on their multivalency (integration of targeting ligands along with other relevant components) [3]. In particular, RNA nanoparticles usually have unique biological histocompatibility, which is crucial to biomedical applications. The size and shape of RNA nanoparticles can be precisely controlled to avoid unwanted toxicity [4,5,6,7], which often appears when polymer-based nanoparticles aggregate to form lager supermolecular complexes [8,9]. Moreover, RNA nanoparticles are largely nonimmunogenic because of the negligible immune response triggered by RNA [10], while protein therapies are more likely to induce antibodies so that targeted cells can develop resistance through repeated administration [11,12]. Thus, RNA nanoparticles are feasible for treating chronic disease repeatedly. Despite these advantages, there are still several challenges for the development of RNA nanotechnology [11,12,13,14,15]. For example, conventional RNA nanoparticles may be dissociated at low concentrations after inhalation [16]. Hence, study of the stability of RNA molecules and nanoparticles and the underlying mechanisms is of vital significance to the development and the biomedical applications of RNA nanotechnology.

Recently, a three-way junction packaging RNA (3WJ-pRNA) has been exploited for its characteristics of self-assembly and thermodynamic stability [17,18,19]. 3WJ-pRNA is the 3WJ part of bacteriophage φ29 pRNA and connects the ATPase-binding region and two interlocking loops of the pRNA [20,21,22,23]. This three-way structure can serve as a scaffold to construct multi-module RNA nanoparticles with functionalities by fusing different ligands to its arms [17,24,25]. Encouragingly, 3WJ-pRNA’s ultrahigh stability in many aspects has been characterized by several studies. Dan Shu et al. demonstrated that RNA nanoparticles based on 3WJ-pRNA can resist high-concentration urea denaturation [19]. In our previous work, the superior structural robustness of 3WJ-pRNA to withstand stretching along its coaxial helices was found. Such extraordinary stability of 3WJ-pRNA can be largely attributed to the existence of Mg^2+^ ions, which can considerably enhance the coupling of the helices of 3WJ-pRNA [26]. Additionally, Mg^2+^ ions can compensate the negative charges in the RNA backbone and reduce electrostatic repulsion, which can also improve the structural stability of 3WJ-pRNA [27]. Hence, the addition of Mg^2+^ ions can result in a low apparent equilibrium dissociation constant (*K_D_*) for the 3WJ-pRNA structure (about 11.4 nm) [28,29]. On the other hand, Na^+^ ions at physiological concentrations (~150 mM) have little effect on RNA stability: *T*_m_ of an RNA helix in 150 mM NaCl solution (48 °C) is close to the case of 0 mM NaCl solution (45 °C) [30]. Hence, it is of practical importance to further study additional aspects of its stability and to understand the underlying mechanisms in a systematic way.

Here, we evaluated the thermal stability of the 3WJ-pRNA in the φ29 motor using all-atom MD simulation. We prove that the 3WJ-pRNA exhibits structural stability and can maintain its original structure in the absence of Mg^2+^ ions at room temperature. We further investigated the behaviors of 3WJ-pRNA under higher temperature condition. The Mg-bound 3WJ-pRNA still remains in its initial three-way junction structure, while the Mg-free 3WJ-pRNA undergoes dramatic structural changes, including irregular distortion of the junction. The contribution of Mg^2+^ ions can be largely attributed to the protective effect of two Mg clamps on the hydrogen bond and base stacking interactions in helices. Taken together, our results reveal the superior thermal stability of 3WJ-pRNA which can be regulated by Mg^2+^ ions. Comprehensive depictions of the thermal stability of pRNA and the underlying regulation mechanisms are helpful for the further development of controllable RNA nanoparticle drug delivery platforms.

## 2. Materials and Methods

As mentioned above, the 3WJ-pRNA is the 3WJ domain of bacteriophage φ29 pRNA and connects the ATPase-binding region and two interlocking loops of the pRNA. The structure of the 3WJ-pRNA was extracted from PDB entry 4KZ2 [31]. In this crystal structure, there are two divalent metal ion-binding sites, which were discovered using Mn^2+^ ions in experiments [31]. Actually, the same coordination sites are found to be occupied by physiologically relevant Mg^2+^ ions as those for Mn^2+^ ions [31]. Thus, we constructed configurations of the Mg-bound and Mg-free 3WJ-pRNA for use in simulations.

We prepared the simulation systems as follows. Each 3WJ-pRNA was embedded in the center of a 10 nm cubic box. The systems were solvated with the TIP3P water [32]. Sodium and chloride ions were added to neutralize the systems and set the total ionic strength to 100 mM and 200 mM. The effective concentration of Mg^2+^ ions in the system of Mg-bound 3WJ-pRNA is about 4 mM, similar to the experimental setting [22]. The systems underwent energy minimization until the maximum force was less than 1000.0 kJ/(mol·nm). Then, both systems were successively equilibrated under a 1-ns NVT ensemble and 2-ns NPT ensemble. Finally, three independent 200-ns simulations at room temperature (*T* = 300 K) and high temperature (*T* = 400 K) were performed for both Mg-bound and Mg-free 3WJ-pRNA. We also performed the simulations of both Mg-bound and Mg-free 3WJ-pRNA at *T* = 370 K with longer time (*t* = 300 ns).

The AMBER14 force field and the Mg^2+^ ions model developed by Allner et al. were selected in simulation runs [33,34,35,36,37,38]. The LINCS algorithm was used to constrain the bond vibrations and the time step was set to 2 fs [39]. Short-range electrostatic and van der Waals interactions were cut off beyond 1.0 nm, whereas the particle-mesh Ewald method was used for long-range electrostatic force [40,41]. The periodic boundary conditions were adopted in all directions [42]. All simulations were carried out with the GROMACS 5.1 simulation package [43]. The simulation snapshots were rendered with VMD 1.9.3 [44]. The electrostatic potentials of the bases in 3WJ-pRNA were calculated by Chimera 1.16 with Adaptive Poisson–Boltzmann Solver (APBS) calculation (Appendix A) [45].

## 3. Results and Discussion

### 3.1. RMSD and Conformation Behaviors at Room Temperature

The Mg-bound 3WJ-pRNA is composed of three helices and a junction region. Helix 1 (H1) is formed by 1U-8G in Chain A and 16A-9C in Chain C binding successively; helix 2 (H2) by 10G-18C in Chain A and 9C-1C in Chain B; helix 3 (H3) by 13G-20C in Chain B and 8U-1G in Chain C (Figure 1A). H1 and H3 are stacked coaxially and two pairs of bases [3G (Chain A) and 7A (Chain C); 4C (Chain A) and 6A (Chain C)] bind Mg^2+^ ions to form two adjacent Mg clamps (Appendix A) [31]. The Mg clamps are parallel to the axis of H1–H3 and thus connect H1 and H3 [31]. The junction region of 3WJ-pRNA is made up of four unpaired uracil bases, including 9U in Chain A and 10U, 11U, and 12U (UUU bulge) in Chain B [31]. Such asymmetric structure between Chain A and Chain B make the RNA strands twist at the junction region and towards Chain B. Thus, the H2 perpendicular to coaxial H1–H3 and 3WJ-pRNA generally presents a T-shaped structure (Figure 1B) [46]. 

Taken together, four unpaired bases at the junction, together with the surrounding base pairs in helices (including the bases involved in the formation of Mg clamps), are the central part of 3WJ-pRNA. Its structural stability is crucial to maintaining relative orientations of three helices and the whole three-way junction structure. Therefore, it serves as the core region of the three-way junction pRNA (Table 1). Thus, the stability of the core region, especially the stability of the junction, is discussed in detailed. There are 6, 2 and 5 base pairs in the three helices, respectively, of the core region (Table 1), according to the rules of Watson–Crick base-pairing (i.e., adenine binding to uracil and cytosine binding to guanine) and wobble base-pairing (e.g., guanine binding to uracil) [47,48]. The corresponding numbers of hydrogen bonds are 16, 5, and 11 [47,48].

We first studied the Mg-bound and Mg-free 3WJ-pRNA at room temperature (*T* = 300 K). Three independent simulations were performed in both systems. A representative trajectory was selected to be studied in detail. The root mean square deviation (RMSD) of the core region was calculated and shown in Figure 1C. In both systems, the RMSDs were maintained below 4 Å in the second half of the simulation (after t = 100 ns) and the RMSD with Mg^2+^ ions was relatively smaller, suggesting that 3WJ-pRNA, at least the core region, can largely maintain its initial structure under room temperature and exhibits great structural stability.

We further investigated the relative orientations of various helices. The final conformations of the core region in both Mg-bound and Mg-free 3WJ-pRNA are shown in Figure 1D. It can be seen that H1 and H3 in both systems maintain a double helix structure and they are still coaxial. Moreover, the relative positions of the four unpaired bases in the junction remain essentially the same and H2 is broadly perpendicular to H1–H3. That is to say, the three-way junction structure has been maintained unchanged. Similar phenomena can be found in the simulations of Mg-bound and Mg-free 3WJ-pRNA in 200 mM NaCl solution (Appendix A). For both systems, the 3WJ structures are largely retained throughout the simulation. Thus, 3WJ-pRNA can roughly maintain its original structure at room temperature even in the absence of Mg^2+^ ions. It can be largely attributed to hydrogen bonds between base pairs and the base stacking between adjacent bases.

### 3.2. RMSD and Conformation Behaviors at High Temperature

We further studied the effect of Mg clamps on the stability of 3WJ-pRNA at a higher temperature (i.e., *T* = 400 K). In the case of the Mg-bound 3WJ-pRNA, the RMSD of the core region remained at less than 5 Å (Figure 2A), indicating that the structure of the core region is still stable. As shown in representative configurations (Figure 2B and Appendix A), the double helix structures of H1 and H3 remain largely unchanged and Mg^2+^ ions still anchor H1 and H3, keeping them in the coaxial state. Although the junction region consisting of four unpaired uracil bases undergoes some degree of deformation, their relative positions change modestly. Moreover, also attributed to the Mg clamps which sustain the coaxial structure of H1–H3 is the stabilizing of the relative positions between U9 and the UUU bulge. As for H2, which is farther from Mg^2+^ ions, it undergoes a certain degree of unwinding, while its overall orientation is similar to the initial structure, i.e., H2 remains perpendicular to H1–H3. Therefore, the core region as well as the whole 3WJ-pRNA still maintains a three-way junction structure. In other words, 3WJ-pRNA exhibits considerable thermal stability in the presence of Mg^2+^ ions, which is essential for its biological applications (e.g., serving as a platform for building various multifunctional and controllable nanoparticles). Similar phenomena about the impacts of Mg^2+^ were also observed in the simulations of Mg-bound 3WJ-pRNA in the solution with different concentration of Na^+^ ions. Mg-bound 3WJ-pRNA in 200 mM NaCl solution exhibits similar structural stability at *T* = 400 K (Appendix A).

As for the case of the Mg-free pRNA, the RMSD of the core region increases rapidly from the beginning of the simulation, exceeding 6 Å at t = 15 ns and further increasing to ~8 Å at t = 75 ns. In the absence of Mg^2+^ ions, the structure of the core region undergoes a dramatic change at a high temperature (*T* = 400 K). Figure 2B (right panel) and Appendix A show the final configuration of the core region in the Mg-free 3WJ-pRNA. It can be observed that both H1 and H3 go through a significant unwinding and their separation increases evidently, which means the coaxial structure has largely disappeared. The unwinding of H2 is even more profound. Moreover, the four unpaired bases show irregular distortion and H2 is no longer perpendicular to H1–H3. Hence, the Mg-free 3WJ-pRNA almost loses its native structure—the T-shaped three-way junction and adopts an irregular Y-shaped structure instead. In the case of 200 mM NaCl solution, the structure of Mg-free 3WJ-pRNA similarly undergoes considerable change at *T* = 400 K (Appendix A). The increase in the concentration of NaCl does not improve the thermal stability of 3WJ-pRNA in the absence of Mg. The conformation change of 3WJ-pRNA may lead to the loss its biological activity and it is not able to act as a scaffold to construct multi-module RNA nanoparticles with functionalities.

In summary, our results show that the structure and relative orientations of three helices in the Mg-bound 3WJ-pRNA remain largely unchanged at the high temperature. As for the case of Mg-free 3WJ-pRNA, it undergoes dramatic structural change, including unwinding of helices and irregular distortion of the junction. The impact of Mg^2+^ ions on the thermal stability of 3WJ-pRNA was also studied at *T* = 370 K with longer simulation times. Similarly, the Mg-bound 3WJ-pRNA largely retains its initial structure, while the Mg-free 3WJ-pRNA undergoes considerable unfolding (Appendix A). In other words, two Mg^2+^ ions can regulate the structural stability of 3WJ-pRNA, which may be instructive for the development of controllable drug delivery platforms based on 3WJ-pRNA.

### 3.3. Interaction Analysis of the Core Region

We further studied the interactions of the bases in the core region under high temperature condition (*T* = 400 K) to discuss the contribution of Mg^2+^ ions to the thermal stability of 3WJ-pRNA. The interactions of the bases are largely attributed to the hydrogen bonds between the base pairs of the complementary strands and the base stacking between adjacent bases. The number of hydrogen bonds in H1–H3 of the core region were calculated and shown in Figure 3A,B. For Mg-bound 3WJ-pRNA, the number of hydrogen bonds in H1 and H3 remain almost unchanged during the whole simulation and the average numbers of H-bonds are 16.2 and 10.3, respectively (based on the last 50 ns trajectory), close to the initial values (i.e., 16 and 11, respectively). In the case of Mg-free 3WJ-pRNA, the loss of the hydrogen bond in H1 is profound: the average number of hydrogen bonds is only 12.7. The number of H-bonds in H3 largely remains constant.

The change of H-bonds in H1 were further discussed in detail by analyzing the hydrogen bonds of each base pair. The changes mainly occur in the base pairs of 14C-3G and 10A-7U and the corresponding numbers are shown in Figure 3C,D. In the case of Mg-bound 3WJ-pRNA, the number of hydrogen bonds in both base pairs remain constant at 3.1 (14C-3G) and 1.8 (10A-7U), on average (based on the last 50 ns trajectory). As for the Mg-free pRNA, the corresponding numbers are 0.0 (14C-3G) and 1.4 (10A-7U). The presence of Mg^2+^ ions not only facilitates the preservation of the hydrogen bonds in the base pair involved in the formation of Mg-clamp (14C-3G) but also improves the stability of the farther base pair (10A-7U).

Overall, the coordination of Mg^2+^ ions might affect the interactions of the bases in the core region and thus regulate the thermal stability of the whole 3WJ-pRNA. Mg^2+^ ions can remarkably stabilize both the structures and relative orientations of various helices by anchoring H1 and H3 through the Mg clamps. In addition to the base pair involved in the formation of the Mg clamp (i.e., 14C-3G), Mg^2+^ ions can also considerably improve the stability of the hydrogen bonds of the farther base pair in H1 (i.e., 10A-7U). Taken together, the base stacking interactions and hydrogen bond interactions of Mg-bound 3WJ-pRNA can be effectively coupled, resulting in the enhanced thermal stability of the RNA structure.

The change of H-bonds in H1 were further discussed in detail by analyzing the hydrogen bonds of each base pair. The changes mainly occur in the base pairs of 14C-3G and 10A-7U and the corresponding numbers are shown in Figure 3C,D. In the case of Mg-bound 3WJ-pRNA, the number of hydrogen bonds in both base pairs remains constant at 3.1 (14C-3G) and 1.8 (10A-7U), on average (based on the last 50 ns trajectory). As for the Mg-free pRNA, the corresponding numbers are 0.0 (14C-3G) and 1.4 (10A-7U). The presence of Mg^2+^ ions not only facilitates the preservation of the hydrogen bonds in the base pair involved in the formation of Mg-clamp (14C-3G) but also improves the stability of the farther base pair (10A-7U).

Overall, the coordination of Mg^2+^ ions might affect the interactions of the bases in the core region and thus regulate the thermal stability of the whole 3WJ-pRNA. Mg^2+^ ions can remarkably stabilize both the structures and relative orientations of various helices by anchoring H1 and H3 through the Mg clamps. In addition to the base pair involved in the formation of the Mg clamp (i.e., 14C-3G), Mg^2+^ ions can also considerably improve the stability of the hydrogen bonds of the farther base pair in H1 (i.e., 10A-7U). Taken together, the base stacking interactions and hydrogen bond interactions of Mg-bound 3WJ-pRNA can be effectively coupled, resulting in the enhanced thermal stability of the RNA structure.

## 4. Conclusions

In this work, we studied the thermal properties of bacteriophage φ29 3WJ-pRNA using all-atom MD simulation. While 3WJ-pRNA can remain its original structure without Mg^2+^ ions at room temperature (*T* = 300 K), only Mg-bound 3WJ-pRNA still maintains its initial three-way junction structure at a higher temperature (*T* = 400 K). The Mg-free 3WJ-pRNA undergoes dramatic deformation under the high-temperature condition. The impacts of Mg^2+^ ions on the structural stability of 3WJ-pRNA can be largely attributed to the protective effect of two Mg clamps on the hydrogen bond and base stacking interactions in helices. Hence, the extraordinary thermal stability of 3WJ-pRNA can be regulated by Mg^2+^ ions.

We also studied the effect of varying NaCl concentrations on the thermal stability of 3WJ-pRNA. Our results show that the structures of Mg-free 3WJ-pRNA in both 100 mM and 200 mM NaCl solutions undergo considerable change at *T* = 400 K. In both cases, Na^+^ ions can occasionally appear at the binding sites of Mg^2+^ ions, while these Na^+^ ions diffuse away quickly and cannot bind with 3WJ-pRNA stably (Appendix A). Thus, Na^+^ ions at such modest concentrations can barely improve the stability of 3WJ-pRNA, which is consistent with the experimental results for the structural stability of RNA molecules in the solution with similar NaCl concentrations. Additionally, we studied the possible binding of Ca^2+^ ions to 3WJ-pRNA in our previous work. Ca^2+^ ions cannot form a “Ca-clamp” with 3WJ-pRNA, and the addition of Ca^2+^ should not improve the thermal stability of 3WJ-pRNA [26].

During the past decade, diverse controllable and multifunctional RNA nanoparticles have been developed for various applications [2]. As revealed by our findings, the 3WJ-pRNA may exhibit the potential to act as a platform to construct multi-module RNA nanoparticles with functionalities, especially in clinical practice. For example, 3WJ-pRNA can serve as a nanocage to harbor various drugs and transport them into cells without being affected by the complex in vivo environments. When adding a chelating agent (e.g., EDTA) and heating the required areas, Mg^2+^ ions can be removed and the structure of the Mg-free 3WJ-pRNA nanocage may thus undergo dramatic changes which result in the disassembly of RNA nanoparticle and controllable drug release.

Along with the addition of Mg^2+^ ions, chemical modifications (e.g., 2’-F modified RNA) can improve the structural stability of RNA molecules [1]. In comparison, the binding of Mg^2+^ ions can be easily regulated. The preparation of Mg-bound 3WJ-pRNA is more convenient [29]. Taken together, the Mg-bound 3WJ-pRNA is suitable for the construction of controllable RNA nanoparticle drug delivery platforms. In addition, there are experiments using both ionic and chemical modifications to further improve the structural stability of RNA molecules [2].

## Figures and Tables

**Figure 1 pharmaceutics-14-01413-f001:**
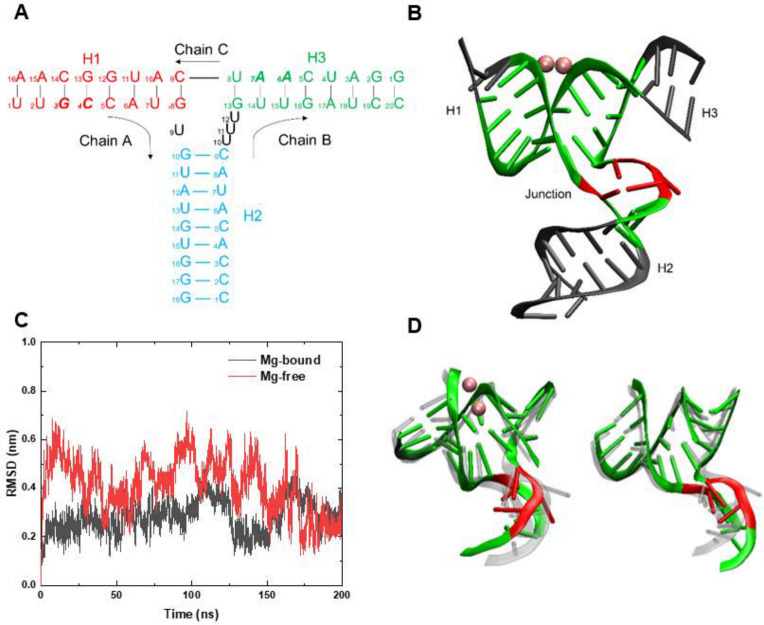
Schematic illustration of the 3WJ-pRNA and the structural characterizations of the core region in both systems at *T* = 300 K. (**A**) The 3WJ domain of pRNA. Three individual RNA strands of the 3WJ are shown as Chain A, Chain B, and Chain C with their trends. The bases are marked with sequential numbers. The helical segments are designated as H1 (red), H2 (blue), and H3 (green), respectively. The core region is defined as four unpaired bases in black and several base pairs around the junction. The bases forming the Mg-clamps [3G (H1) and 7A (H3); 4C (H1) and 6A (H3)] are marked in bold. (**B**) Schematic representative of the Mg-bound 3WJ-pRNA with different parts in corresponding colors. (**C**) The RMSDs of the core region in Mg-bound (black) and Mg-free (red) 3WJ-pRNA with respect to the initial structure. (**D**) Representative snapshots of the core region in Mg-bound (left panel) and Mg-free (right panel) 3WJ-pRNA. The initial structures are shown as transparent, and the final structures are shown in normal colors.

**Figure 2 pharmaceutics-14-01413-f002:**
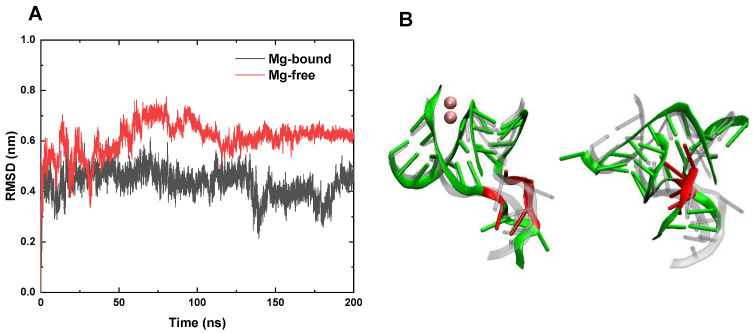
The structural characterizations of the core region in both systems at *T* = 400 K. (**A**) The RMSDs of the core region in Mg-bound (black) and Mg-free (red) 3WJ-pRNA with respect to the initial structure. (**B**) Representative snapshots of the core region in the Mg-bound (left panel) and the Mg-free (right panel) 3WJ-pRNA. The initial structures are shown as transparent, and the final structures are shown in normal color.

**Figure 3 pharmaceutics-14-01413-f003:**
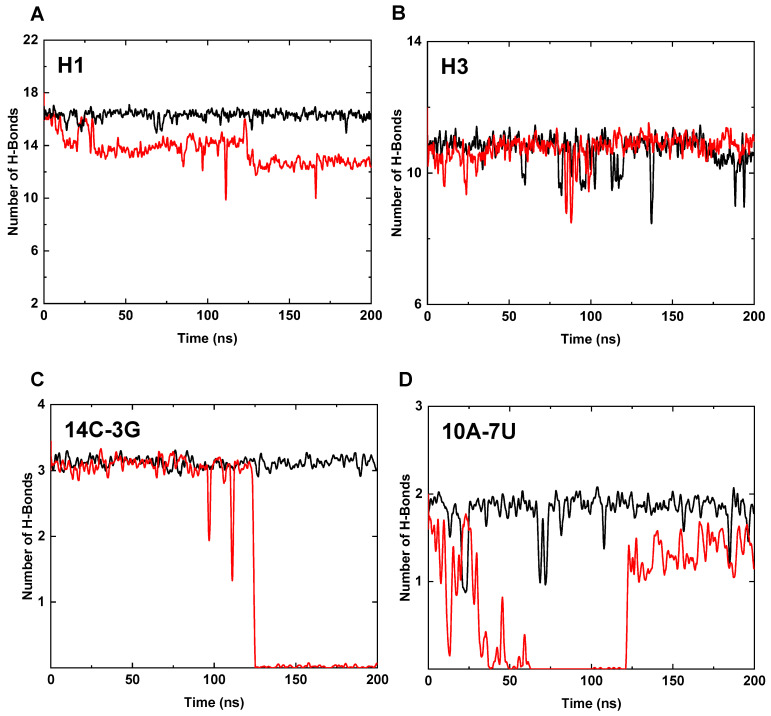
The hydrogen bonds in various helices at *T* = 400 K. (**A**) The number of hydrogen bonds in H1 (Mg-bound in black, Mg-free in red, the same below). (**B**) The number of hydrogen bonds in H3. (**C**) The number of hydrogen bonds between 14C-3G of H1. (**D**) The number of hydrogen bonds between 10A-7U of H1.

**Table 1 pharmaceutics-14-01413-t001:** Sequence of the core region of 3WJ-pRNA.

Area	Type and Label of Nucleobases	Hydrogen Bonds
**Junction**	Chain A: 9U Chain C: 10U 11U 12U	0
**Helix 1**	Chain A: 13G 14C 15C 16A 17U 8G Chain C: 14C 13G 12C 11U 10A 9C	16
**Helix 2**	Chain A: 10G 11U Chain B: 19C 18A	5
**Helix 3**	Chain B: 13G 14U 15U 16G 17A Chain C: 18U 17A 16A 15C 14U	11

## Data Availability

The data that support the findings of our study are available from the corresponding author upon reasonable request.

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
