# Peer review of "Mg2+ Ions Regulating 3WJ-PRNA to Construct Controllable RNA Nanoparticle Drug Delivery Platforms"

_pharmaceutics, 2022, doi:10.3390/pharmaceutics14071413_

Round 1

Reviewer 1 Report

The article (pharmaceutics-1738143) provides certain fundamental results for improving the stability of 3WJ-pRNA, and to my opinion, it could be accepted for the publication after considering the following comments:     

- The authors determine the effect of Mg2+ on DNA atomic electrostatic potentials, and changes related to the interbase pair parameters.

- Authors should compare their Mg2+ modified structure with a control modification (e.g, 2′F-modified, LNA), and describe the advantages/disadvantages.

- The authors should determine the apparent equilibrium dissociation constant (Kd) for both modified and unmodified structure.

Reviewer 2 Report

However, it has long been known that the structural integrity and biological activity of RNA depend on the type and concentration of counterions in solution. The strong electrostatic repulsion between closely packed backbone phosphate anions tends to unwind the RNA. Divalent cations such as magnesium play a critical role in reducing this repulsion and thus in stabilizing the conformation of the folded RNA. Therefore, in the introduction, the author should consider the effect of monovalent and divalent salts on the stability of RNA molecules. Simulation should be carried out at several concentrations of monovalent salt and varying the concentration of Mn ions. When discussing the results, it was necessary to compare the results obtained in the simulation with experimental data on the effect of monovalent salts and the concentration of Mg ions on the stability of RNA molecules.

Reviewer 3 Report

In "Mg2+ ions Regulating 3WJ-pRNA to Construct Controllable RNA Nanoparticle Drug Delivery Platforms", Drs Li and Chen present the simulation of RNA nanoparticle assemblies using the three-way junction packaging. The simulation and the conclusions are correct. However, the significance of the work is relatively low, as the authors just applied already available methods to identify the stabilization effects of Mg2+. Moreover, the stabilization effects of Mg2+ at comparable RNA positions are already known (RNA 9, 1226-37, 2013), and may be correlated with the structure of 3WJ-pRNA. Thus, I cannot recommend this study for publication in it current form. I recommend the authors to provide more insights and further scope to the study by including the following simulations.

1. Study the effect of several different temperatures. 

2. Check other ions besides Mg2+. Is there another ion that can provide better binding and stabilization than Mg2+?

Round 2

Reviewer 1 Report

Accept

Reviewer 2 Report

The manuscript has been sufficiently improved and can be published.

Reviewer 3 Report

The authors have properly revised their work.